# pomBseen: An automated pipeline for analysis of fission yeast images

**Makoto Ohira** , **Nicholas Rhind** *

Department of Biochemistry and Molecular Biotechnology, University of Massachusetts Medical School, Worcester, Massachusetts, United States of America

* nick.rhind@umassmed.edu

## Abstract

Fission yeast is a model organism widely used for studies of eukaryotic cell biology. As such, it is subject to bright-field and fluorescent microscopy. Manual analysis of such data can be laborious and subjective. Therefore, we have developed pomBseen, an image analysis pipeline for the quantitation of fission yeast micrographs containing a bright-field channel and up to two fluorescent channels. It accepts a wide range of image formats and produces a table with the size and total and nuclear fluorescent intensities of the cells in the image. Benchmarking of the pipeline against manually annotated datasets demonstrates that it reliably segments cells and acquires their image parameters. Written in MATLAB, pomBseen is also available as a standalone application.

## Introduction

The fission yeast *Schizosaccharomyces pombe* is a powerful eukaryotic model system. It has been used to dissect fundamental concepts in nuclear and cytoplasmic cell biology [1]. A key tool for the study of yeast cell biology is wide-field microscopy, which allows for the analysis of cell size and, using fluorescently tagged proteins, the levels of cytoplasmic and nuclear proteins. Microscopy data provides crucial insight in to the regulation of protein expression and localization. Quantitation of such data is straight forward using image-analysis software, such as ImageJ [2]. However, manual quantitation of many cells, which is often required to obtain statistically significant results, can be tedious and time consuming. Moreover, it introduces subjectiveness into the analysis.

A number of labs have addressed these issues by developing automated image analysis programs that can identify cells in micrographs and extract cell size and fluorescent-intensity parameters. A number of programs were designed specifically for pombe [3–6], whereas others were designed to be more generally applicable [7–14]. However, we found that none of the packages available in 2019, when we started this project, provided a simple, robust solution for the quantitation of our fluorescent images. In particular, we found that packages designed for mammalian-cell images performed poorly on fission yeast.

We developed pomBseen, an automated analysis pipeline, to measure the length and fluorescence intensity of pombe cells and their nuclei. We chose to use the high-level

**Data Availability Statement:** All code, a users' manual and example files are available at GitHub <https://github.com/makotojohira/pomBseen>.

**Funding:** This work was funded by NIH grant GM134300 to NR. The funders had no role in study

design, data collection and analysis, decision to publish, or preparation of the manuscript.

**Competing interests:** The authors have declared that no competing interests exist.

programming language MATLAB, because it has a well-developed arsenal of image-analysis functions, is widely used and is well-supported.

## Results and discussion

pomBseen is a MATLAB program for the automated analysis of fluorescent micrographs of fission yeast. The input is an image file, with a bright-field image in the first channel, zero, one or two fluorescent images in the subsequent channels, and, if included, the imbedded pixels-to-micron ratio metadata. Image data is extracted and passed, along with the metadata, to the analysis routines of pomBseen. pomBseen identifies individual cells and records their size and, if present, fluorescence from the whole cell and, if a nucleus is identifiable, from the nucleus. The automatically saved outputs of pomBseen are a composite image of segmented cells and nuclei and a CSV file containing analysis results (Tables 1 and S1). The optionally saved outputs of pomBseen are figures representing the major functional steps of the program which give the user insights to the processing steps.

Fig 1 illustrates the overall flow of data through pomBseen (see Methods for details). The main steps are segmenting the bright-field image to identify cells, measuring the size of the identified cells, segmenting the fluorescent image(s) to identify nuclei, measuring the size of the identified nuclei, and measuring the fluorescent signal in each cell and nucleus. pomBseen generally runs without the need for user input, but does have one quality control step at which the user can exclude cells from analysis. pomBseen reports an image or plot from most of the main steps of the analysis (Fig 1A), which the user may save at their discretion. These figures are automatically deleted upon the next run of pomBseen.

The data is output in a CSV file with 11 columns for two-channel image files and 23 columns for three-channel files. The output content is listed in Table 1 and an example generated from Fig 1 is presented as S1 Table.

The first column shows the index number of the cell, corresponding to the numbers reported in the final figure with superimposed nucleus and cell (see Fig 1, panel 12). The data in a given row belongs to the cell whose number is reported in column 1 of the same row. This

**Table 1. pomBseen output data.**

| Output data | Image channel | Mask channel |
|---|---|---|
| Cell index | 1 | - |
| Number of nuclei per cell | 2 | 2 |
| Cell length | 1 | - |
| Nucleus area | 2 | 2 |
| Cell area | 1 | - |
| Mean nucleus intensity | 2 | 2 |
| Cell width | 1 | - |
| Whole cell intensity | 2 | 1 |
| Background intensity (bg) | 2 | - |
| Nucleus intensity—bg | 2 | 2 |
| Whole cell intensity—bg | 2 | 1 |
| Mean nucleus intensity | 3 | 2 |
| Nucleus area | 3 | 3 |
| Mean nucleus intensity | 3 | 3 |
| Mean nucleus intensity | 2 | 3 |
| Whole cell intensity | 3 | 1 |
| Background intensity | 3 | - |
| Nucleus intensity—bg | 3 | 3 |
| Whole cell intensity—bg | 3 | 1 |
| Nucleus intensity—bg | 3 | 2 |
| Nucleus intensity—bg | 2 | 3 |
| Number of nuclei per cell | 3 | 3 |

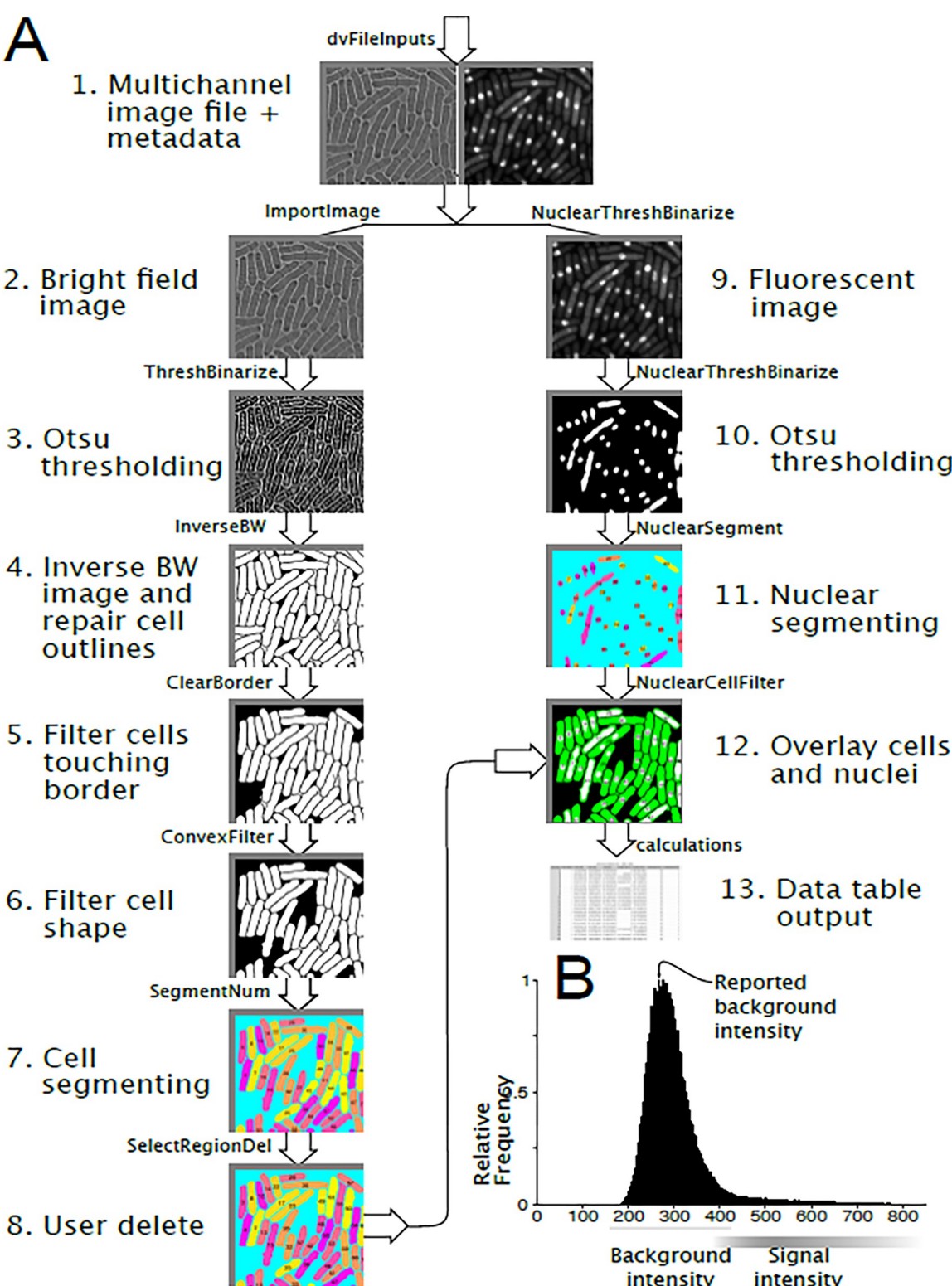

**Fig 1. pomBseen workflow. A**) pomBseen workflow. Images shown are the upper left quadrant of the analyzed images. **B**) Background estimation for fluorescent channels. See Methods for details.

final superimposed figure (or two figures for three-channel DV files) are saved by pomBseen for the user's reference.

The second column reports the number of nuclei segmented in each cell. Most cells should have a single nucleus per cell, but post-mitotic cells will have two nuclei. pomBseen should successfully segment multiple nuclei if the fluorescent protein is highly expressed and evenly distributed throughout the nucleus and minimally in the cytoplasm (see Fig 1, for example cell numbers 23 and 26).

The third column reports the major axis length of the cell, the fourth reports the nuclear area, the fifth the cell area, the sixth the mean nuclear intensity, the seventh the cell width, the eight the whole cell intensity, and so on (Table 1).

The header for each column shows a numerical prefix followed by a colon and the title of the data in the column. The numerical prefix refers to the channel from which the data is generated. For example, in the first column, the cell index always derives from the bright-field image in channel 1. The number of nuclei in the second column is culled from the nuclear fluorescence image in channel 2.

Several columns have two numbers arranged like a fraction. For example, the title of the 6th column is: *2/2*: *Mean Nuc Int*. The first number denotes the channel from which mean nuclear intensity is calculated. The second number denotes the channel by which the nucleus was segmented. So, the 6th column title refers to: mean nuclear intensity of nuclei imaged in channel 2, and segmented from channel 2.

The reason for this approach is that nuclei may not be well-segmented in both fluorescent channels. pomBseen calculates and reports each channel's mean nuclear fluorescence using nuclei segmented from each channel, for a total of four permutations of data. The user can select the combination which is most appropriate.

Sometimes, pomBseen is unable to segment nuclei for some fluorescently labeled proteins (perhaps due to very low nuclear expression or excessive cytoplasmic concentration). For this reason, we also report whole cell fluorescence intensity. In this case, masking is done using the more reliable bright-field image of the whole cell in channel 1.

Because of the stringent filtering, false positive segments are very rare. In routine datasets of hundreds of cells, we find no segments that do not correspond to cell images. The converse effect of this stringent filtering is that the false negative rate can be high, especially for poor-quality image. In high-quality images with well-dispersed cells, false negatives are not a problem. For instance, for the image in Fig 2C–2E, the false negative rate is between 4 and 15%. However, for images of poor quality or ones in which there is significant cell overlap, it can be over 90%.

Validation of pomBseen is shown in Fig 2, illustrating the similarity of the output cell length and nuclear fluorescence data compared to manual measurements made using ImageJ ($r2 = 0.95$–$0.99$ for length, Fig 2A and 2C–2E, and $r2 = 0.98$ for fluorescence, Fig 2B). Length variability was affected by several outliers, three of which are indicated in the figure. pomBseen assigned some background to these cell segments, thereby artifactually increasing their length. The user can remove such erroneous segmentations at the quality-control step. An example of such a removal is shown in Fig 1 (Cell 42, Steps 7 and 8). Fig 2C–2F show that pomBseen segmentation is robust over a 7-fold range of sizes. We used a strain that expresses Wee1 under the control of a β-estradiol-sensitive promoter, allowing us to produce isogenic asynchronous populations with a range of cell lengths [15].

Fluorescence variability was also small. However, there are a number of cells where the fluorescence value recorded by pomBseen is zero (blue data points in Fig 2B). These are cells in which pomBseen was unable to successfully segment the nucleus and which were therefore

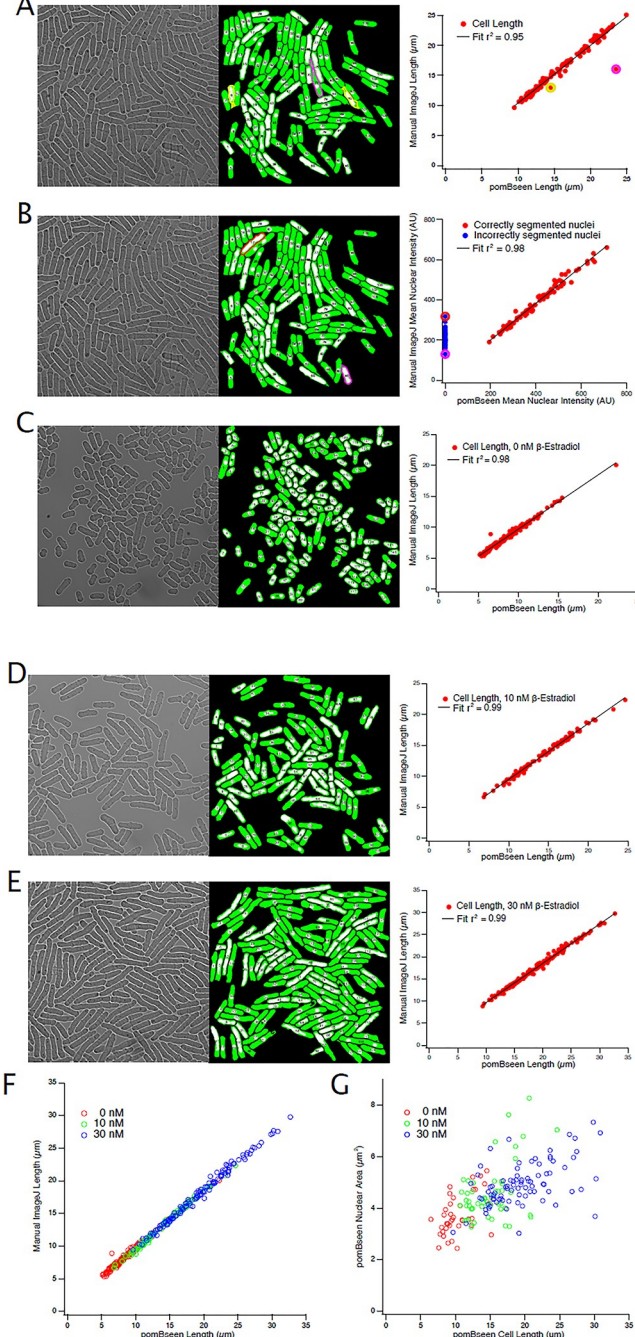

**Fig 2. pomBseen validation.** Validation of pomBseen quantitation by comparison to manually quantitated cells. A) Comparison of cell length measurements of asynchronous yFS1136 cells. The left panel is a bright-field image of the cells; the middle panel is an overlay of the segmented cells and nuclei; and the right panel is a comparison of the pomBseen and manual measurements of the cells. Cells incorrectly segmented by pomBseen are indicated in yellow and magenta in the middle and right panels. Manual length measurements were made with the ImageJ Line Selection tool. B) Comparison of nuclear fluorescence intensity in the same cells as in A. Nuclei for which the intensity was too low to be segmented properly by pomBseen are plotted in blue and two are outlined in red or magenta in the middle and right panels. Nuclei were manually selected and measured with the ImageJ Freehand Selection tool. C-E) Comparison of cell length measurements of asynchronous yFS1078 cells grown in different concentrations of β-estradiol, as indicated in the legend of the right panel. F) A collation of the data from C-E. G) Nuclear area plotted against cell length, showing that nuclear size increases with cell size, as expected [16].

excluded from analysis (see Methods for details on nuclear filtering). Two of these outliers are indicated in the Fig 2B.

Fig 2G shows that the size of the segmented nuclei increases with cell length, as expected [16].

For pomBseen to segment cells, the bright-field image must be slightly defocused to obtain a narrow, bright halo around each cell (Fig 1). It is crucial that the data be acquired with such halos. New users often require one or two rounds of data collection and analysis to appreciate the importance of the bright-field image quality. Therefore, it is recommended that new users attempt a pilot analysis with a well-behaved control strain before attempting to collect experimental data.

Although pomBseen was written specifically for DeltaVision fluorescence microscopes and uses its manufacturer-specific data formats, pomBseen should be compatible with a wide range of open access and proprietary image formats, some of which will also include calibration data for the images. If such metadata is not included, pomBseen will report results in pixels.

pomBseen, a users' manual, standalone versions of pomBseen, and example files are available at GitHub <https://github.com/makotojohira/pomBseen>.

## Methods

pomBseen is a MATLAB analysis pipeline that runs on MATLAB 2019a and compatible releases and requires the Image Analysis and Bio-Formats libraries. It takes as input any image data file supported by Bio-Formats <https://docs.openmicroscopy.org/bio-formats/5.3.4/supported-formats.html>. pomBseen has been extensively tested with DV files, the proprietary image file type produced by DeltaVision microscopes, but should work with over 100 other supported proprietary and open-source file formats. In particular, we have confirmed that it works with TIFF files. It also uses a set of predefined parameters that are stored in the `Parameters.m` file (Table 2). These parameters should not need to be modified, but can be if a particular need arises.

**Table 2. pomBseen parameters.**

| pomBseen Function | Subroutine | MATLAB Parameter | pomBseen Parameter Variable | Value |
|---|---|---|---|---|
| ThreshBinarize | adaptthresh | sensitivity | ThreshBinSensitivity | 0.55 |
| | | Neighborhoodsize | ThreshBinNeighbrhd | [15 15] |
| InverseBW | bwareaopen | max pixels | InverseBWMaxPix | 600 |
| ClearBorder | imclearborder | connectivity | ClearBorderConn | 8 |
| | bwareaopen | max pixels | ClearBorderMaxPix | 1200 |
| SegmentNum | bwconncomp | connectivity | SegmentNumConn | 4 |
| | Mask area | Area Min | SegmentNumAreaMin | 500 |
| | Mask area | Area Max | SegmentNumAreaMax | 100000 |
| ConvexFilter | Filter | | ConvexFilterSlope | 12.8571 |
| | Filter | | ConvexFilterIntercept | 12.5 |
| NuclearThreshBinarize | imopen | neighborhood | NucThreshBinNeighbrhd | ones(5,5) |
| | bwareaopen | max pixels | NucThreshBinMaxPix | 200 |
| NuclearSegment | bwconncomp | connectivity | NucSegmentConn | 4 |
| | mask | NucArea | NucSegmentMaskMax | <155 |
| NuclearCellFilter | bwconncomp | connectivity | NucCellFilterConn | 4 |
| | NCRatioIndex | | NucCellFilterNCRatioMin | > 0.2 |

pomBseen can be run within the MATLAB environment or as a standalone application compiled for either MacOS or Windows operating systems. Code, a users' manual and example files are available at GitHub <https://github.com/makotojohira/pomBseen>.

The four images analyzed here were taken of yFS1136 (h- leu1::pFS531 (cdc25-NeonGreen leu1) ura4-D18 wee1::pWAU-50 (P(adh1):wee1-50ts) or yFS1078 (h- leu1::pFS461(P(adh1): ZEV leu1) ura4-D18 ade6-216 P(ZE3V):wee1 (kanMX) cdc25-NeonGreen (hygMX) nph6-mCherry (ura4)) and cells [17] grown in YES medium to mid log phase. To obtain different sized cells for the analysis in Fig 2C–2E, cells were grown in presence of different levels of beta-estradiol [15], as indicated in the figure legend. Cells were fixed in 100% methanol at -80˚C. Cells were rehydrated in 1X PBS and imaged by widefield fluorescence microscopy with a DeltaVision-enabled Olympus IX71 inverted microscope with a 60x/1.42 oil-immersion objective. These images are representative of the data for which we developed pomBseen.

## Workflow

The flow of pomBseen is described as follows and is illustrated in Fig 1A. The section numbers correspond to the panel numbers in the figure.

1. Import data file and extract image and metadata from each channel
   `dvFileInputs` imports data files using the bfmatlab package of Bio-Formats <https://docs.openmicroscopy.org/bio-formats/5.3.4/users/matlab/index.html>. pomBseen uses the bfmatlab `bfopen` function to open any Bio-Formats-supported user-selected image file. Once the file is open, standard MATLAB functions extract the image channels and the metadata from the original data file.
   A pomBseen function later in the workflow, `NuclearCellFilter`, extracts the pixel and physical size calibration data. That calibration data is applied to the pomBseen output calculations, which are made in pixel units, and converts them to units of micron (for length) or micron squared (for area). If the input files do not contain such metadata, which is the case for TIFF files, data is output in units of pixels and can be converted after the analysis.

2. Extract bright-field image and sharpen the focus
   `ImportImage` extracts the first channel bright-field image data. It then sharpens the grayscale bright-field image to improve the thresholding and segmenting functions in the subsequent steps.

3. Apply Otsu's thresholding method and convert the gray-scale bright-field image to a black-and-white image
   The first major step in the pomBseen workflow is to segment individual cells in the bright-field image. Traditionally, the first step in segmenting is to assign an object with pixel values of 1 (white) while the background is assigned pixel values of 0 (black). What we expect to see is a translation of the gray-scale image into white objects on a black background. Successful segmentation is dependent on the quality of the image. Machine learning is a powerful solution to variable image quality [7,10], but relies on training sets that contain images of comparable quality. We opted for a simpler solution of asking the user to defocus the image slightly resulting in a clear bright halo surrounding each cell. The pomBseen function `ThreshBinarize` identifies the halo by using Otsu's thresholding algorithm [18], which is embedded in the MATLAB function `imbinarize`. `imbinarize` can be modified with various parameters to assist in thresholding (Table 2). Successful thresholding in pomBseen results in halos which are assigned a value of 1 and everything else including the cells a value of 0.

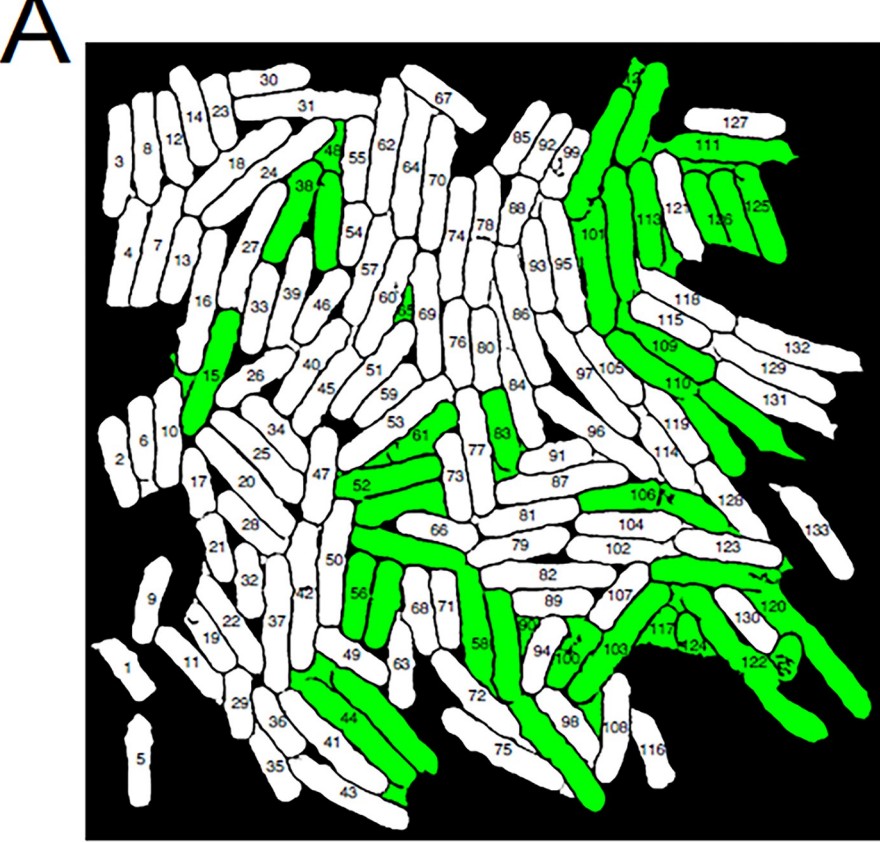

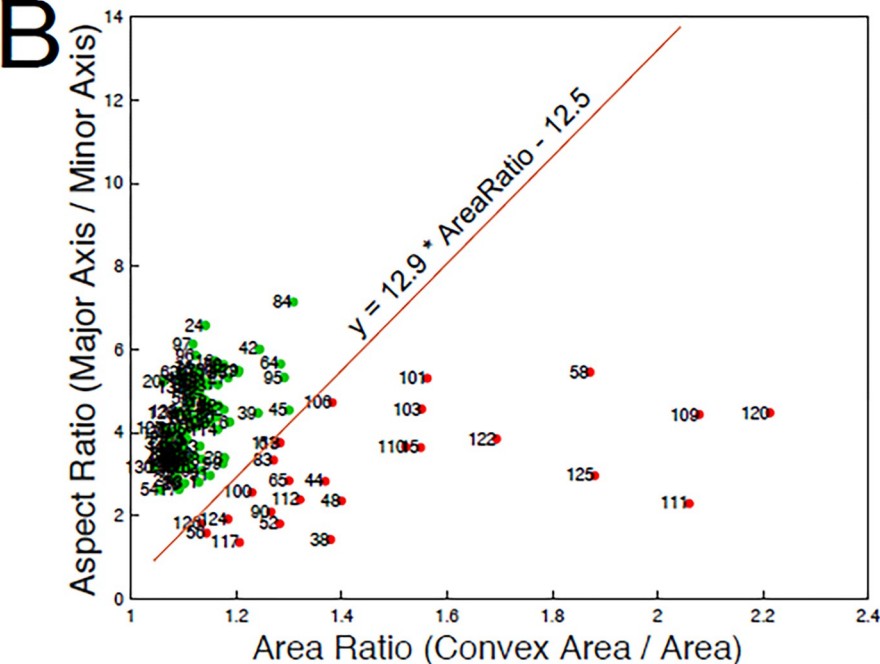

**Fig 3. pomBseen convexity filter.** A Convexity filter is applied to remove mis-segmented cells. A) Properly segmented cells are shown in white and improperly segmented cells and background regions are shown in green. B) The AspectRatio v. AreaRatio of these segmented regions, with properly segmented cells in green and improperly segmented cells and background regions in red are plotted. The line `y = 12.9 * AreaRatio - 12.5`, which separates them, is shown. See Methods for details.

4. Invert the BW image and repair cell outlines
Our goal is to segment cells, not their halos, so we need to reverse the assignment of 1's and 0's. pomBseen function `InverseBW` reverses the thresholding output of `ThreshBinarize` such that cells have a value of 1 and the halo and most of the background has a value of 0. Microscopy images typically have significant intensity variability throughout the image. Therefore, halos are often discontinuous and thresholded cells can have holes in them. `Threshbinarize` calls a standard MATLAB function, `bwareaopen`, to correct some of these smaller defects. Each of the MATLAB functions called by pomBseen are modified with parameters, the most important of which are listed and thus easily modified in the pomBseen file `Parameters.m`.

5. Eliminate cells touching the border
Cells contacting the image border cannot usually be accurately measured for length. MATLAB function `imclearborder` removes objects touching the border and eliminates potentially truncated cells from analysis. The pomBseen function `ClearBorder` uses `imclearborder` and `bwareaopen` again to clean up the segmented image.
5b. Segment and number cells
The pomBseen function `SegmentNum` makes a first pass at counting segmented cells. During this operation, objects less than 500 pixels and larger than 100,000 pixels are eliminated because such objects are not likely to be cells.

6. Apply a convexity filter to eliminate any shapes which are not likely to be pombe cells
The pomBseen function `ConvexFilter` removes improperly segmented cells and background regions which remain despite previous filters. These regions are readily distinguished by their irregular concave shape which is distinct from a pombe cell. Fission yeast cells are convex and have a large aspect ratio. These geometric characteristics are captured in AspectRatio, the ratio of major axis length to minor axis length, and AreaRatio, the ratio of convex area (the area of the smallest convex region containing the object) to actual area of the object. These parameters were plotted and properly segmented cells were observed to have a distinct linear relationship between AspectRatio and AreaRatio. Objects above the line `y = 12.9 * AreaRatio - 12.5` were most often properly segmented cells, and those below that cutoff were most often improperly segmented cells or background artifacts (Fig 3), which are thereby automatically filtered out. The parameters defining the line separating cells from artifacts can be modified in the `Parameters.m` file, in case other datasets have different distributions of cells when plotted in AspectRatio v. AreaRatio space.

7. Re-segment and re-number cells
After the preceding filtering steps, cells are re-segmented and re-numbered to remove excluded objects from the cell index list.

8. Allow user to select cells for deletion
`SelectRegionDel` gives the user the option to select segmented regions or cells for deletion. A few artifacts (most often a fusion between a cell and a small background region) may continue to linger in the image field despite various filtering operations. Therefore, an option was introduced here to allow the user to manually select and eliminate any segmented object(s) from analysis.

9. Extract fluorescent image and calculate background
Many of the same operations are conducted on the fluorescent channel(s) in `NuclearThreshBinarize` as on the bright-field channel. One additional operation for the fluorescent channel, however, is the calculation of the background fluorescent

intensity. This is done by determining the most frequent intensity value on an intensity histogram of the image (Fig 1B).

10. Apply Otsu's thresholding to nuclei
A key difference between bright-field and fluorescent channels is that for fluorescently-labeled proteins that are strongly expressed and uniformly distributed through the nucleus, segmenting nuclei is much more straightforward. There is no need for indirect methods like defocusing for a halo and inverting the resulting thresholded image. However, since cytoplasmic background fluorescence varies significantly from cell to cell, it is not practical to invoke a single value by which to threshold and segment all of the nuclei in the image. Instead, `NuclearThreshBinarize` masks the image for each cell, then thresholds the nucleus (or sometimes two nuclei) for only that selected cell. This loop is repeated for all the segmented cells in the image.

11. Segment and number nuclei
The nuclei are segmented and numbered much as the cells are from the bright-field image. In `NuclearSegment` the masking filter is different from that applied to cells since nuclei are significantly smaller and less variable than cells.

12. Filter nuclei based on nucleus:cell area ratio
Segmented nuclei that fail a threshold nucleus:cell area ratio are rejected in `NuclearCellFilter`. If the intensity of the nucleus fails to rise significantly above the cell's fluorescence, pomBseen is unable to threshold, and therefore to segment a distinct nucleus, and a significant amount of the cell area is erroneously included in the segmentation of the nucleus. Since the nucleus:cell area ratio is normally less than 20% [16], if that ratio exceeds 20%, pomBseen discards the segmented region, and the mean intensity is reported as zero. The cell is not deleted, since other means of quantifying fluorescence, such as whole cell fluorescence, can be used instead.
12b. If there is a second fluorescent channel, repeat the above analysis for channel 3, starting at step 9

13. Calculate output data
The final step is to record nuclear and whole cell fluorescence, cell length, and background intensity, and save the data in a CSV file. Cell length is calculated as the major axis length of the segmented cell. Nuclear intensity was calculated as the mean intensity within the segmented nucleus. Sometimes, nuclear intensity for a given channel is not significantly distinct from the cytoplasm such that a nucleus can be segmented. If a nucleus can be segmented in a different channel, that segmented region can be used to define the area in the channel with the unsegmented nucleus in which a mean intensity can be calculated. Whole cell intensity is calculated as the mean intensity within the segmented cell. The index number of each nucleus is matched with the index of the cell in which it resided, and so these values are reported in the same row, given by the index.

## Potential modifications

**Changing channel assignments.** As currently configured, pomBseen requires that the image file contains the bright-field image as channel 1. It is straight forward to modify pomBseen to use the files with the bright-field data in another channel.

The pomBseen function `dvFileInputs` assigns the input data file, which may contain multiple image channels and associated metadata, to a variable simply called R. The function

`dvFileInputs` counts the number of channels of data used in R1, then extracts the images using a for loop. The images are saved and named in the order of the channel number.

The next pomBseen function, `ImportImage`, then retrieves the first channel data R1{1,1} assuming it is a bright-field image. The next channel, which is assumed to be fluorescent, is R1 {2,1}. The third channel is R1{3,1}.

However, if a user for some reason must make a different channel contain the bright-field or fluorescent images, it is straightforward to reassign the variables accordingly.

**Using other image file formats.** pomBseen was designed to analyze DeltaVision DV image files. However, the Bio-Formats tool used to open DV files works with over 100 other proprietary and open-source image formats <https://docs.openmicroscopy.org/bio-formats/5. 3.4/supported-formats.html>. So, pomBseen should work with almost all image formats. In particular, we have confirmed that it works with TIFF files.

**Changing parameters.** The parameters for various functions were assigned to optimize pomBseen performance in segmenting cells. These optimal values could conceivably change with different users or imaging systems or a number of other variables. Most parameters are listed in the file `Parameters.m`, such that the user may easily find them and modify them as needed.

**Changing the number of figures outputted during pomBseen.** The current version of pomBseen outputs one or more figures during the execution of each function. These are intended to verify the output of each function and as a reference for the user if needed. The user may opt to save one or more of these figures. They are all closed upon the next run of pomBseen. The user may close them manually by typing `close all` into the command window (at the bottom of the MATLAB user interface).

The user may also modify pomBseen, by commenting out the lines which control the output of each figure. In each function, the user may find a few lines similar to the following:

```
figure('Numbertitle', 'off','Name','Function: NuclearFilter.
m Overlay');
imshow(overlay);
Typing a % in front of each line will turn it from an
executable line of code into a comment, and thus disable it
(without deleting it-so commenting is an easily reversible
action). The above lines commented out would look as follows:
%figure('Numbertitle', 'off','Name','Function:
NuclearFilter.m Overlay');
%imshow(overlay);
```

## Supporting information

**S1 Table. pomBseen output.**
(CSV)

## Acknowledgments

We are grateful to Samir Bashir, Wendy Kam and other members of the Rhind lab for help with development and testing of pomBseen.

## Author Contributions

**Conceptualization:** Makoto Ohira, Nicholas Rhind.

**Funding acquisition:** Nicholas Rhind.

**Investigation:** Makoto Ohira.

**Methodology:** Makoto Ohira.

**Project administration:** Nicholas Rhind.

**Resources:** Nicholas Rhind.

**Software:** Makoto Ohira.

**Supervision:** Nicholas Rhind.

**Validation:** Nicholas Rhind.

**Visualization:** Makoto Ohira, Nicholas Rhind.

**Writing – original draft:** Makoto Ohira, Nicholas Rhind.

**Writing – review & editing:** Makoto Ohira, Nicholas Rhind.

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
