## [Decision Letter · Decision Letter 0]

27 Jan 2023

PONE-D-23-00477pomBseen: An Automated Pipeline for Analysis of Fission Yeast ImagesPLOS ONE

Dear Dr. Rhind,

Thank you for submitting your manuscript to PLOS ONE. After careful consideration, we feel that it has merit but does not fully meet PLOS ONE’s publication criteria as it currently stands. Therefore, we invite you to submit a revised version of the manuscript that addresses the points raised during the review process.

The main issues with the manuscript raised by Reviewer 3 were to provide more detailed and systematic assessment of the method’s performance over a range of images, so I encourage the authors to focus on assessment of the proposed method.

We look forward to receiving your revised manuscript.

Kind regards,

Khan Bahadar Khan, Ph.D

Academic Editor

PLOS ONE

Journal Requirements:

"We are grateful to Samir Bashir, Wendy Kam and other members of the Rhind lab for help with 

development and testing of pomBseen. This work was funded by NIH grant GM134300 to NR."

Reviewers' comments:

Reviewer's Responses to Questions

**Comments to the Author**

1. Is the manuscript technically sound, and do the data support the conclusions?

Reviewer #1: Partly

Reviewer #2: Yes

Reviewer #3: Partly

2. Has the statistical analysis been performed appropriately and rigorously? 

Reviewer #1: N/A

Reviewer #2: Yes

Reviewer #3: Yes

3. Have the authors made all data underlying the findings in their manuscript fully available?

Reviewer #1: Yes

Reviewer #2: No

Reviewer #3: Yes

4. Is the manuscript presented in an intelligible fashion and written in standard English?

Reviewer #1: Yes

Reviewer #2: Yes

Reviewer #3: Yes

5. Review Comments to the Author

Reviewer #1: The authors of “pomBseen: An Automated Pipeline for Analysis of Fission Yeast Images” present a software tool written in MATLAB to perform basic, but important and often time-consuming, steps when analysing fission yeast microscopy images. These steps include cell segmentation and quantification of the nuclear signal (if present). I think the method is well explained and I appreciated the detailed user manual the authors wrote.

However, it is not clear to what extent this software is an improvement of existing methods. I think it is still a valuable contribution because it’s always good for the end user to have multiple options.

To make it clearer to the reader on it can contribute to the image analysis, the authors should address three main points:

1. A proof-of-concept biological experiment is missing. The authors should try to replicate some existing measurements to show that indeed pomBseen can be used to obtain biological insights. A simple experiment that I can think of is to show nuclear size scaling with cell volume. Also, why is it important to compute the fluorescence intensity of the nuclear signal? Does it correlate with some biological information?

2. A benchmark of the segmentation performance against state-of-the-art algorithms. This benchmark should include a comparison to the latest convolutional neural networks available. I can suggest YeaZ and Cellpose for the bright-field, and StarDist and Cellpose for the fluorescence signal. For a benchmark example see the YIT benchmark and MOTA metrics http://yeast-image-toolkit.biosim.eu/pmwiki.php.

3. To make it more accessible to the reader, the authors should expand the text with more information. Specifically, I propose the following changes:

a. Abstract should start with a couple of sentences to describe the application field, why fission yeast is a good model

organisms and what kind of problem there is that pomBseen is solving

b. Introduction should start with a description of the biological importance of fission yeast as a model organism and why

it is important to study the nucleus using fluorescence microscopy (and what are the advantages of this technique)

c. In the introduction, the authors cite some of the existing software, but some other important ones are missing.

Additionally, don’t just add the references, but write the name of the software. It makes it easy for the reader to

know what you are writing about, without searching in the references. Please cite the following papers:

• StarDist (DOI: https://doi.org/10.1007/978-3-030-00934-2_30 )

• Cellpose (DOI: https://doi.org/10.1038/s41592-020-01018-x )

• Cell-ACDC (DOI: https://doi.org/10.1186/s12915-022-01372-6 )

• The algorithms used in the YIT benchmark (http://yeast-image-toolkit.biosim.eu/pmwiki.php)

Minor comments:

- Writing “brightfield phase-contrast image” is wrong. Either it is phase-contrast OR brightfield. Maybe the authors wanted to write “wide-field” instead of “bright-field”? On the other hand, the images look bright field to me. Please explain.

- In Figure 1, it is not clear how the image after thresholding (step 3.) can be processed to obtain image shown in step 4. Please add that there is a fill holes step

- In Figure 1 after ClearBorder step, why are some cells touching the bottom and right border still there? Is this a crop of the top-left part of a larger image? Please specify.

- In Figure 2 it is not clear how the intensities are calculated in Fiji. Are there masks generated by pomBseen used in Fiji or are the cells segmented manually? Are those mean intensities? Please specify.

- Please separate Figure 2 in panel A and B.

Reviewer #2: This is fine study carried out by authors. However, few comments can further improve the quality of this work.

1. The abstract of article need to be refine. (1) Clearly mention the problem faced in this filed (2) What is your contribution and (3) how this method reliable for fission yeast micrographs than standard other methods available?

2. The dataset used for this experiment must be referenced and explain how many images part of your experiment?

3. Convexity filter to remove miss-segmented cells explanation needed more how it help to increase your segmentation results.

4. The figure 2 and 3 have multiple figures. Split each figures into figure 2(a) and figure 2(b) etc. similar for figure 3. Moreover, explain each figure in text separately.

Reviewer #3: The authors propose “pomBseen” as a new image analysis pipeline for fission yeast micrographs capable of identifying cells and extracting cell sizes and fluorescent-intensity parameters. The method takes in user-selected images and metadata as inputs, and it produces a CSV file output containing results that are indexed by cell and ready for analysis. The method is implemented in MATLAB and combines functions from existing image processing libraries into a user-friendly program. The method’s automatic segmentation and measurement methods are highly correlated with manual measurements, and the user is provided opportunity to supervise the method discard problematic cells.

In my assessment, pomBseen offers a substantial improvement over existing image analysis pipelines for fission yeast micrographs in several ways. The method achieves close correspondence with human measurements, and the implementation is accessible and flexible. That said, there are a few ways in which the manuscript could be improved prior to publication:

Major issues:

- The underlying data and examples that are used to validate the proposed method are not clearly described in the manuscript. You display results for two different image of manually segmented cells (Fig 2 and Fig 3), but it is unclear whether these images were specially chosen for illustrative purposes, or whether they are representative of the method’s performance more broadly. I ask that you include a more detailed and systematic assessment of the method’s performance over a range of images that are representative of scenarios where others might use the method.

- The observed correlations between length measurements from pomBseen and human measurement are compelling, but these results could be better assessed in the context of other object identification applications if the authors were to report the intersection over union (IOU) of the automatically segmented cells compared to the manually.

- From the example and information given, it’s unclear what the false positive and true positive rates of the proposed method are. You briefly describe filtering techniques intended to remove objects whose make them unlikely to be pombe cells, but it isn’t clear at what rate other false positives might occur. Even though users having the opportunity to manually remove erroneous results, it seems that the efficacy of the proposed method to obtain a sample that is equivalent to manual segmentation depends upon the rate at false positives occur and need to be manually excluded.

Minor issues:

- It’s common for machine learning approaches, such as this one, to work best for the data and conditions they were developed under. In this regard, you mention “Although pomBseen was written specifically for DeltaVision fluorescence microscopes and uses its manufacturer-specific data formats, pomBseen should be compatible with a wide range of open access and proprietary image formats, some of which will also include calibration data for the images”. I’m curious if you’ve investigated this claim by applying the proposed method to images obtained by other devices or in different formats. If so, it would be worthwhile to provide an indication of how performance compares.

- Similar to the previous point, in Step 6 of the methods you describe an equation for filtering out anomalous cells using aspect ratios. Were the parameters of this equation estimated using your data? Additional information would be helpful if others wanted to adjust these defaults to better calibrate pomBseen (as needed) for their own applications.

- The caption for Figure 3 is brief and the methods section where it is referenced doesn’t fully describe the two components of the figure.

- There are several paragraphs in the results and discussion describing components of the output CSV file. I felt like these seemed out of place, and that information could be better presented in a table, figure, or example.

6. PLOS authors have the option to publish the peer review history of their article (what does this mean?). If published, this will include your full peer review and any attached files.

Reviewer #1: No

Reviewer #2: No

Reviewer #3: No

---

## [Author Response · Author response to Decision Letter 0]

18 Feb 2023

Reviewer #1: The authors of “pomBseen: An Automated Pipeline for Analysis of Fission Yeast Images” present a software tool written in MATLAB to perform basic, but important and often time-consuming, steps when analysing fission yeast microscopy images. These steps include cell segmentation and quantification of the nuclear signal (if present). I think the method is well explained and I appreciated the detailed user manual the authors wrote.

However, it is not clear to what extent this software is an improvement of existing methods. I think it is still a valuable contribution because it’s always good for the end user to have multiple options.

To make it clearer to the reader on it can contribute to the image analysis, the authors should address three main points:

1. A proof-of-concept biological experiment is missing. The authors should try to replicate some existing measurements to show that indeed pomBseen can be used to obtain biological insights. A simple experiment that I can think of is to show nuclear size scaling with cell volume.

The requested analysis is now included in Figure 2G.

Also, why is it important to compute the fluorescence intensity of the nuclear signal? Does it correlate with some biological information?

One purpose of pomBseen is to measure fluorescence intensity as a proxy for fluorescently-tagged protein concentration. Thus, we report both nuclear and total cellular fluorescence from all fluorescent channels.

2. A benchmark of the segmentation performance against state-of-the-art algorithms. This benchmark should include a comparison to the latest convolutional neural networks available. I can suggest YeaZ and Cellpose for the bright-field, and StarDist and Cellpose for the fluorescence signal. For a benchmark example see the YIT benchmark and MOTA metrics http://yeast-image-toolkit.biosim.eu/pmwiki.php.

The proposed analysis is not relevant to our work. Our approach is benchmarked against manual annotation and preforms well. We make no claims about performance relative to other computational approaches.

3. To make it more accessible to the reader, the authors should expand the text with more information. Specifically, I propose the following changes:

a. Abstract should start with a couple of sentences to describe the application field, why fission yeast is a good model organisms and what kind of problem there is that pomBseen is solving

Done.

b. Introduction should start with a description of the biological importance of fission yeast as a model organism and why it is important to study the nucleus using fluorescence microscopy (and what are the advantages of this technique)

Done.

c. In the introduction, the authors cite some of the existing software, but some other important ones are missing.

The additional software references have been added, but they are not listed by name. This manuscript as not meant as review of currently available segmentation approaches.

Additionally, don’t just add the references, but write the name of the software. It makes it easy for the reader to know what you are writing about, without searching in the references. Please cite the following papers:

• StarDist (DOI: https://doi.org/10.1007/978-3-030-00934-2_30 )

• Cellpose (DOI: https://doi.org/10.1038/s41592-020-01018-x )

• Cell-ACDC (DOI: https://doi.org/10.1186/s12915-022-01372-6 )

• The algorithms used in the YIT benchmark (http://yeast-image-toolkit.biosim.eu/pmwiki.php)

Minor comments:

- Writing “brightfield phase-contrast image” is wrong. Either it is phase-contrast OR brightfield. Maybe the authors wanted to write “wide-field” instead of “bright-field”? On the other hand, the images look bright field to me. Please explain.

The images are bright-field and are now described as such.

- In Figure 1, it is not clear how the image after thresholding (step 3.) can be processed to obtain image shown in step 4. Please add that there is a fill holes step

Done.

- In Figure 1 after ClearBorder step, why are some cells touching the bottom and right border still there? Is this a crop of the top-left part of a larger image? Please specify.

Yes, the displayed images are cropped from the top-left of the full images. This manipulation is now explained.

- In Figure 2 it is not clear how the intensities are calculated in Fiji. Are there masks generated by pomBseen used in Fiji or are the cells segmented manually? Are those mean intensities? Please specify.

The cells were manually segmented and mean intensities are reported, as is now stated in the figure legend.

- Please separate Figure 2 in panel A and B.

Done.

Reviewer #2: This is fine study carried out by authors. However, few comments can further improve the quality of this work.

1. The abstract of article need to be refine. (1) Clearly mention the problem faced in this filed (2) What is your contribution and (3) how this method reliable for fission yeast micrographs than standard other methods available?

Done.

2. The dataset used for this experiment must be referenced and explain how many images part of your experiment?

Done.

3. Convexity filter to remove miss-segmented cells explanation needed more how it help to increase your segmentation results.

The description of how the convexity filter is applied had been modified in section 6 of the methods and the line separating properly segmented cells from improperly segmented cells and background regions has been labeled in Figure 3B.

4. The figure 2 and 3 have multiple figures. Split each figures into figure 2(a) and figure 2(b) etc. similar for figure 3. Moreover, explain each figure in text separately.

Done.

Reviewer #3: The authors propose “pomBseen” as a new image analysis pipeline for fission yeast micrographs capable of identifying cells and extracting cell sizes and fluorescent-intensity parameters. The method takes in user-selected images and metadata as inputs, and it produces a CSV file output containing results that are indexed by cell and ready for analysis. The method is implemented in MATLAB and combines functions from existing image processing libraries into a user-friendly program. The method’s automatic segmentation and measurement methods are highly correlated with manual measurements, and the user is provided opportunity to supervise the method discard problematic cells.

In my assessment, pomBseen offers a substantial improvement over existing image analysis pipelines for fission yeast micrographs in several ways. The method achieves close correspondence with human measurements, and the implementation is accessible and flexible. That said, there are a few ways in which the manuscript could be improved prior to publication:

Major issues:

- The underlying data and examples that are used to validate the proposed method are not clearly described in the manuscript. You display results for two different image of manually segmented cells (Fig 2 and Fig 3), but it is unclear whether these images were specially chosen for illustrative purposes, or whether they are representative of the method’s performance more broadly. I ask that you include a more detailed and systematic assessment of the method’s performance over a range of images that are representative of scenarios where others might use the method.

We now include three additional images representing a range of cell sizes and describe how the data was acquired and state that they are representative of data for which we designed pomBseen.

- The observed correlations between length measurements from pomBseen and human measurement are compelling, but these results could be better assessed in the context of other object identification applications if the authors were to report the intersection over union (IOU) of the automatically segmented cells compared to the manually.

We did not segment the images manually. We only measured their lengths manually. Therefore, we are unable to report an IOU comparison.

- From the example and information given, it’s unclear what the false positive and true positive rates of the proposed method are. You briefly describe filtering techniques intended to remove objects whose make them unlikely to be pombe cells, but it isn’t clear at what rate other false positives might occur. Even though users having the opportunity to manually remove erroneous results, it seems that the efficacy of the proposed method to obtain a sample that is equivalent to manual segmentation depends upon the rate at false positives occur and need to be manually excluded.

The rate of false positives is very low across a wide variety of images. The rate of false negatives is very sensitive to image quality and can range from as little as 5%, as in Figure 2C-E, to almost 100% in poor-quality images. These considerations are now discussed in the manuscript.

Minor issues:

- It’s common for machine learning approaches, such as this one, to work best for the data and conditions they were developed under. In this regard, you mention “Although pomBseen was written specifically for DeltaVision fluorescence microscopes and uses its manufacturer-specific data formats, pomBseen should be compatible with a wide range of open access and proprietary image formats, some of which will also include calibration data for the images”. I’m curious if you’ve investigated this claim by applying the proposed method to images obtained by other devices or in different formats. If so, it would be worthwhile to provide an indication of how performance compares.

No, we have not.

- Similar to the previous point, in Step 6 of the methods you describe an equation for filtering out anomalous cells using aspect ratios. Were the parameters of this equation estimated using your data? Additional information would be helpful if others wanted to adjust these defaults to better calibrate pomBseen (as needed) for their own applications.

Yes, the parameters were generated from our data, although not the data analyzed in this paper. As now described more explicitly in Section 6 of the Methods, the parameters can be adjusted in the Parameter.m file.

- The caption for Figure 3 is brief and the methods section where it is referenced doesn’t fully describe the two components of the figure.

Both have been expanded.

- There are several paragraphs in the results and discussion describing components of the output CSV file. I felt like these seemed out of place, and that information could be better presented in a table, figure, or example.

An example of the CSV output is now included as Table S1 and is referenced in the text.

---

## [Decision Letter · Decision Letter 1]

9 Mar 2023

PONE-D-23-00477R1pomBseen: An automated pipeline for analysis of Fission Yeast imagesPLOS ONE

Dear Dr. Rhind,

Thank you for submitting your manuscript to PLOS ONE. After careful consideration, we feel that it has merit but does not fully meet PLOS ONE’s publication criteria as it currently stands. Therefore, we invite you to submit a revised version of the manuscript that addresses the points raised during the review process.

Please carefully observe and address the comments of  reviewer 1 on benchmarking your method against other automated tools. And even if comparing to manual segmentation was enough (which is not), the authors only compared the length and not the instance segmentation output of the model.

We look forward to receiving your revised manuscript.

Kind regards,

Khan Bahadar Khan, Ph.D

Academic Editor

PLOS ONE

Reviewers' comments:

Reviewer's Responses to Questions

**Comments to the Author**

1. If the authors have adequately addressed your comments raised in a previous round of review and you feel that this manuscript is now acceptable for publication, you may indicate that here to bypass the “Comments to the Author” section, enter your conflict of interest statement in the “Confidential to Editor” section, and submit your "Accept" recommendation.

Reviewer #1: (No Response)

Reviewer #2: All comments have been addressed

Reviewer #3: (No Response)

2. Is the manuscript technically sound, and do the data support the conclusions?

Reviewer #1: Partly

Reviewer #2: Yes

Reviewer #3: (No Response)

3. Has the statistical analysis been performed appropriately and rigorously? 

Reviewer #1: No

Reviewer #2: Yes

Reviewer #3: (No Response)

4. Have the authors made all data underlying the findings in their manuscript fully available?

Reviewer #1: Yes

Reviewer #2: Yes

Reviewer #3: (No Response)

5. Is the manuscript presented in an intelligible fashion and written in standard English?

Reviewer #1: Yes

Reviewer #2: Yes

Reviewer #3: (No Response)

6. Review Comments to the Author

Reviewer #1: I thank the authors for having addressed my comments, I think the manuscript has improved.

However, I don't think they can skip benchmarking their method against other automated tools. And even if comparing to manual segmentation was enough (which is not), the authors only compared the length and not the instance segmentation output of the model.

This is a standard in the field and without it, the reader cannot properly assess the performance of the model.

Additionally, there are some discrepancies:

1. You mentioned that the "cells were manually segmented", but then you also mentioned that you did not segment the images manually. Even if you did not segment the cells manually, the IoU comparison cannot be skipped.

2. Related to 1, you mentioned that the "Manual ImageJ Intensity" of Figure 2 are mean intensities, but how do you get mean intensities if you did not segment them manually? The length is not enough to get the mean intensities.

3. If I am not mistaken, the Figure 2 legend still doesn't say that the intensities are mean intensities. Please add this to the axis label on the figures too.

4. The reason why you cannot skip benchmarking against other automated methods is that manual analysis can be subjective, as you properly mentioned in the abstract. Therefore, comparing only to manual analysis does not address this manual analysis bias.

Reviewer #2: (No Response)

Reviewer #3: (No Response)

7. PLOS authors have the option to publish the peer review history of their article (what does this mean?). If published, this will include your full peer review and any attached files.

Reviewer #1: No

Reviewer #2: No

Reviewer #3: No

---

## [Author Response · Author response to Decision Letter 1]

4 Apr 2023

The major concern of Review 1 is that we have not compared our method to other automated image-analysis methods. I concede that such a comparison would be a useful exercise, however that is not the goal of our work. Our purpose in creating and publishing pomBseen is to provide a tool for the analysis of our image data that is as good as our current standard of analysis, which is manual measurement using ImageJ. I am confident that we have done so.

Direct responses to the reviewer's specific comments are below.

Reviewer #1: I thank the authors for having addressed my comments, I think the manuscript has improved.

However, I don't think they can skip benchmarking their method against other automated tools. And even if comparing to manual segmentation was enough (which is not), the authors only compared the length and not the instance segmentation output of the model.

I have addressed the issue of benchmarking above. I do not understand the point that "the authors only compared the length and not the instance segmentation output of the model". Our approach measures two parameters: cell length and nuclear fluorescence. As, described below, in Figure 2 we compared both: length as manually measured using the ImageJ Line Selection tool and mean nuclear intensity as manually measured using the Freehand Selection tool. So, although we did not manually segment the cells, per se, we did validate the both segmentation steps of pomBseen by comparing them to manual measurements.

This is a standard in the field and without it, the reader cannot properly assess the performance of the model.

Additionally, there are some discrepancies:

1. You mentioned that the "cells were manually segmented", but then you also mentioned that you did not segment the images manually. Even if you did not segment the cells manually, the IoU comparison cannot be skipped.

My statement in my response to the Reviewer that "cells were manually segmented" was careless. In the figure legend, we accurately stated that we "manually quantitated" the cells. As now more fully explained in the figure legend, we manually measured the cell length using the ImageJ Line Selection tool and manually measured the mean intensity of the nuclei using the Freehand Selection tool, neither of which capture the data that would be needed for an IoU analysis.

2. Related to 1, you mentioned that the "Manual ImageJ Intensity" of Figure 2 are mean intensities, but how do you get mean intensities if you did not segment them manually? The length is not enough to get the mean intensities.

We manually measured the mean intensity of the nuclei using the Freehand Selection tool, as now explained in the figure legend.

3. If I am not mistaken, the Figure 2 legend still doesn't say that the intensities are mean intensities. Please add this to the axis label on the figures too.

Done.

4. The reason why you cannot skip benchmarking against other automated methods is that manual analysis can be subjective, as you properly mentioned in the abstract. Therefore, comparing only to manual analysis does not address this manual analysis bias.

We do not aim, nor claim, to be as good as or better than any other automated method. We simply claim to be as good as, but more efficient than, our current method of manual analysis. And I am confident we have robustly demonstrated as much.

---

## [Decision Letter · Decision Letter 2]

19 Apr 2023

PONE-D-23-00477R2pomBseen: An automated pipeline for analysis of Fission Yeast imagesPLOS ONE

Dear Dr. Rhind,

Thank you for submitting your manuscript to PLOS ONE. After careful consideration, we feel that it has merit but does not fully meet PLOS ONE’s publication criteria as it currently stands. Therefore, we invite you to submit a revised version of the manuscript that addresses the points raised during the review process.

It is observed that reviewer 1 comments are not fully incorporated. Benchmarking against other methods cannot be skipped. It's quite simple to do and it takes max one day. Run YeaZ or cellpose, run your model (which does segmentation since it's clearly shown in Figure 1), compute IoU and compare. Please revise manuscript by including benchmarking against other methods.

We look forward to receiving your revised manuscript.

Kind regards,

Khan Bahadar Khan, Ph.D

Academic Editor

PLOS ONE

Journal Requirements:

Reviewers' comments:

Reviewer's Responses to Questions

**Comments to the Author**

1. If the authors have adequately addressed your comments raised in a previous round of review and you feel that this manuscript is now acceptable for publication, you may indicate that here to bypass the “Comments to the Author” section, enter your conflict of interest statement in the “Confidential to Editor” section, and submit your "Accept" recommendation.

Reviewer #1: (No Response)

Reviewer #2: All comments have been addressed

Reviewer #3: (No Response)

2. Is the manuscript technically sound, and do the data support the conclusions?

Reviewer #1: Partly

Reviewer #2: Yes

Reviewer #3: Yes

3. Has the statistical analysis been performed appropriately and rigorously? 

Reviewer #1: N/A

Reviewer #2: Yes

Reviewer #3: Yes

4. Have the authors made all data underlying the findings in their manuscript fully available?

Reviewer #1: Yes

Reviewer #2: No

Reviewer #3: Yes

5. Is the manuscript presented in an intelligible fashion and written in standard English?

Reviewer #1: Yes

Reviewer #2: Yes

Reviewer #3: Yes

6. Review Comments to the Author

Reviewer #1: I have nothing to add. Benchmarking against other methods cannot be skipped. It's quite simple to do and it takes max one day. Run YeaZ or cellpose, run your model (which does segmentation since it's clearly shown in Figure 1), compute IoU and compare.

Reviewer #2: The author(s) of article taken all my comments into careful consideration and have made improvements to the manuscript accordingly. The revised article improved the overall quality of the manuscript.

Reviewer #3: It is my opinion that the work as it currently stands is suitable for publication. While I agree with the other reviewer that benchmarking would be a worthwhile addition, I also believe that the authors have provided enough for future researchers to perform that work if they seek to compare the efficacy of different tools/methods.

7. PLOS authors have the option to publish the peer review history of their article (what does this mean?). If published, this will include your full peer review and any attached files.

Reviewer #1: No

Reviewer #2: No

Reviewer #3: No

---

## [Author Response · Author response to Decision Letter 2]

19 May 2023

We have decided not to perform the bench marking suggested by Reviewer 1.

---

## [Editor Report · Decision Letter 3]

5 Jun 2023

PONE-D-23-00477R3

pomBseen: An automated pipeline for analysis of Fission Yeast images

PLOS ONE

Dear Dr. Rhind,

Thank you for submitting your manuscript to PLOS ONE. After careful consideration, we have decided that your manuscript does not meet our criteria for publication and must therefore be rejected.

I am sorry that we cannot be more positive on this occasion, but hope that you appreciate the reasons for this decision.

Kind regards,

Khan Bahadar Khan, Ph.D

Academic Editor

PLOS ONE

Additional Editor Comments:

Thank you to the reviewer for highlighting a critical concern regarding the benchmarking and evaluation of the proposed method in the manuscript titled "pomBseen: An automated pipeline for analysis of Fission Yeast images." After carefully considering the reviewer's comment, I agree that the manuscript lacks a crucial aspect of performance evaluation, specifically the comparison of the proposed method with other automated tools commonly used in the field.

To ensure that readers can properly assess the performance of the model, it is necessary to include a benchmarking analysis against existing automated methods. Additionally, the comparison should extend beyond just length measurements and should also evaluate the instance segmentation output of the model.

Based on this evaluation, I regret to inform the authors that the manuscript cannot be accepted for publication in its current form. However, if the authors address the concerns raised by including a benchmarking analysis against other automated tools and evaluate the instance segmentation output, I would be open to reconsidering the manuscript for potential acceptance.

I appreciate the authors' efforts in conducting this research and encourage them to make the necessary revisions to strengthen the evaluation and ensure the manuscript's suitability for publication.

- - - - -

---

## [Author Response · Author response to Decision Letter 3]

25 Jun 2023

Reviewer 1: I have nothing to add. Benchmarking against other methods cannot be skipped. It's quite simple to do and it takes max one day. Run YeaZ or cellpose, run your model (which does segmentation since it's clearly shown in Figure 1), compute IoU and compare.

As explained in our previous responses to Reviewer 1, the proposed benchmarking is not relevant to our work. Our approach is benchmarked against the most robust available ground truth, manual annotation, and preforms well. We make no claims about performance relative to other computational approaches.

To be more explicit, there is no possible outcome from the proposed benchmarking analysis that would contribute to the characterization of pomBseen. We demonstrated that pomBseen robustly recapitulates manual annotation (r2= 0.98-0.99). If comparison between pomBseen and other automated pipelines were to show comparable performance, it would confirm that pomBseen works robustly, which has already been demonstrated. If it were to show a discrepancy between pomBseen and other automated pipelines, it would demonstrate that the other pipelines fail to robustly recapitulate manual annotation, which, although perhaps of interest to users of those pipelines, is not relevant to our work.

Reviewer 2: The author(s) of article taken all my comments into careful consideration and have made improvements to the manuscript accordingly. The revised article improved the overall quality of the manuscript.

We appreciate the reviewers acknowledgement of our response to their constructive comments and suggestions.

Reviewer 3: It is my opinion that the work as it currently stands is suitable for publication. While I agree with the other reviewer that benchmarking would be a worthwhile addition, I also believe that the authors have provided enough for future researchers to perform that work if they seek to compare the efficacy of different tools/methods.

We appreciate the reviewers acknowledgement of our response to their constructive comments and suggestions and emphasize their agreement with our contention that the Review 1's request for benchmarking, although a reasonable suggestion that someone might want to do, is beyond what is required for publication in PLoS ONE.

Editor: Thank you to the reviewer for highlighting a critical concern regarding the benchmarking and evaluation of the proposed method in the manuscript titled "pomBseen: An automated pipeline for analysis of Fission Yeast images." After carefully considering the reviewer's comment, I agree that the manuscript lacks a crucial aspect of performance evaluation, specifically the comparison of the proposed method with other automated tools commonly used in the field.

To ensure that readers can properly assess the performance of the model, it is necessary to include a benchmarking analysis against existing automated methods. Additionally, the comparison should extend beyond just length measurements and should also evaluate the instance segmentation output of the model.

Reviewer 1's concern is that we did not benchmark our method against other automated image analysis packages fundamental misunderstanding of the purpose of our work. Our goal in creating and publishing pomBseen is to provide a tool for the analysis of image data that is as good as, but more efficient than, the current standard of analysis, which is manual measurement using ImageJ. I concede that, in the field of computational image analysis, benchmarking against other packages maybe the standard. However, my lab does not work in that field. We work in the field of yeast cell biology, and the standard in our field is manual image analysis, as demonstrated by the fact that we have used that approach in all of our recent work (PMIDs 33683349, 30957637, 28479325). Therefore, pomBseen is a robust and useful tool for work in our field, and we have adequately demonstrated as much in our manuscript.

In particular, I contest the Editors claim that “[t]o ensure that readers can properly assess the performance of the model, it is necessary to include a benchmarking analysis against existing automated methods”. Such benchmarking is not required to assess the performance of pomBseen. What is required in the benchmarking against the ground truth of manual annotation, which we provide.

---

## [Decision Letter · Decision Letter 4]

17 Aug 2023

PONE-D-23-00477R4

pomBseen: An automated pipeline for analysis of Fission Yeast images

PLOS ONE

Dear Dr.Nicholas Rhind,

Thank you for submitting your manuscript to PLOS ONE. After careful consideration, we feel that it has merit but does not fully meet PLOS ONE’s publication criteria as it currently stands. Therefore, we invite you to submit a revised version of the manuscript that addresses the points raised during the review process.

We look forward to receiving your revised manuscript.

Kind regards,

Sudhir Kumar Rai, Ph.D

Academic Editor

PLOS ONE

Journal Requirements:

Additional Editor Comments (if provided):

Being a pombe person, I personally feel the develop tool is extremely important for pombe community. To further test the potential of developed tool, author can also provide some study under different stress conditions which lead to change in dimension as well as nuclear integrity.

Reviewers' comments:

Reviewer's Responses to Questions

**Comments to the Author**

1. If the authors have adequately addressed your comments raised in a previous round of review and you feel that this manuscript is now acceptable for publication, you may indicate that here to bypass the “Comments to the Author” section, enter your conflict of interest statement in the “Confidential to Editor” section, and submit your "Accept" recommendation.

Reviewer #4: (No Response)

Reviewer #5: (No Response)

Reviewer #6: (No Response)

Reviewer #7: (No Response)

2. Is the manuscript technically sound, and do the data support the conclusions?

Reviewer #4: Yes

Reviewer #5: Yes

Reviewer #6: Yes

Reviewer #7: Partly

3. Has the statistical analysis been performed appropriately and rigorously? 

Reviewer #4: Yes

Reviewer #5: Yes

Reviewer #6: Yes

Reviewer #7: I Don't Know

4. Have the authors made all data underlying the findings in their manuscript fully available?

Reviewer #4: Yes

Reviewer #5: Yes

Reviewer #6: Yes

Reviewer #7: Yes

5. Is the manuscript presented in an intelligible fashion and written in standard English?

Reviewer #4: Yes

Reviewer #5: Yes

Reviewer #6: Yes

Reviewer #7: Yes

6. Review Comments to the Author

Reviewer #4: In this manuscript, the authors present an image analysis tool for microscopic images of fission yeast, allowing for automated cell segmentation and quantification of fluorescent signals. As the other reviewer suggested, a comparison with other existing methods would strengthen the manuscript. However, there is no standard method, and many fission yeast labs still perform measurements manually.

I have only a few minor comments.

It would be helpful to provide examples of analysis of images taken with systems other than deltavision to demonstrate the versatility of pomBseen.

What are DV files? Is it a specific file type for the deltavision microscope?

Reviewer #5: I am a new reviewer and has seen the previous reviews and editor comments. In my opinion this paper can be published in PLoS One without a benchmarking and it provides a usedul and valid tool (as validated by comparison with manually analysed data). However, I suggest this sentence in the introduction to be better qualified, in what way the other tools were not robust or simple? Is there a particular experimental set up this tool would be more suitable. Please be as specific as you can be about the tools tried.

"However, we found that none of the packages available when we started this project provided a simple, robust solution for the quantitation of our fluorescent images."

Reviewer #6: Review of Ohira and Rhind Revision 4

This manuscript provides a versatile, efficient and accurate program to automate measurements of micrograph images of fission yeast. I understand some reviewers feel the output should be compared to other software, but I do not agree. The authors have demonstrated in Fig 2 that the output is highly accurate regarding cell length and nuclear intensity. The community of biologists that study fission yeast will find this program to be extremely useful.

Specific comments

1. No page number or line numbers are provided. This makes it hard to describe comments. I will consider the abstract and Intro to be page 1.

2. Page 1, 9 lines from bottom. Add space to correct Tables 1and S1.

3. Page 3, text- pomBseen should be compatible with a wide range of open access…” I would like to see some more specificity here to help the reader understand which microscope systems are compatible. Can the authors provide more details.

Reviewer #7: Briefly, the manuscript described development of an automated analysis pipeline, to measure the length and fluorescence intensity of yeast cells and their nuclei. Using programming language MATLAB for image-analysis which is widely spread and is interesting approach, however, there are some issues to be addressed in this current manuscript. My comments are below.

1. The authors should edit figures and results more comprehensive and precise. Figures seems to be insufficient of explanation of the abbreviation without consideration for readers.

2. The authors should offer a more comprehensive explanation of the improvement points compared to the references presented in the Introduction section. For instance, Baybay EK et al. described the 3D segmentation of radially symmetric cells, which involved the combination of 2D brightfield segmentation with 3D morphological extrapolation using fission yeast (Reference #4). However, the authors did not provide the limitations of this research and a detailed explanation of how their pipeline improves upon the existing approach.

3. In the Method section, the authors descripted about the images analyzed for study. A couple of strains of cells from reference (#17), however, the strains of the reference were for functional examination of the Cdc 13 and Cdc 25 expression. The authors should explain about the reason for choosing the specific strains for their study.

4. Furthermore, the authors should provide the number of image data used for validating PomBseen and more detailed calculating methods, catering to future users rather than programmers.

5. The authors should thoroughly describe the significance of employing the MATLAB program for image analysis in comparison to other methods. This aspect constitutes a key strength of the manuscript.

7. PLOS authors have the option to publish the peer review history of their article (what does this mean?). If published, this will include your full peer review and any attached files.

Reviewer #4: No

Reviewer #5: **Yes: **

Reviewer #6: **Yes: **

Reviewer #7: No

---

## [Author Response · Author response to Decision Letter 4]

23 Aug 2023

Reviewer #4:

In this manuscript, the authors present an image analysis tool for microscopic images of fission yeast, allowing for automated cell segmentation and quantification of fluorescent signals. As the other reviewer suggested, a comparison with other existing methods would strengthen the manuscript. However, there is no standard method, and many fission yeast labs still perform measurements manually.

I have only a few minor comments.

It would be helpful to provide examples of analysis of images taken with systems other than deltavision to demonstrate the versatility of pomBseen.

We do not have access to another comparable microscope system. However, we have confirmed that pomBseen works with TIFF files. Therefore, pomBseen should work with all systems that can save images in TIFF format. We have now explicitly stated the general applicability of pomBseen in the manuscript.

What are DV files? Is it a specific file type for the deltavision microscope?

DV files are the proprietary image file type produced by DeltaVision microscopes.

Reviewer #5:

I am a new reviewer and has seen the previous reviews and editor comments. In my opinion this paper can be published in PLoS One without a benchmarking and it provides a usedul and valid tool (as validated by comparison with manually analysed data). However, I suggest this sentence in the introduction to be better qualified, in what way the other tools were not robust or simple? Is there a particular experimental set up this tool would be more suitable. Please be as specific as you can be about the tools tried.

"However, we found that none of the packages available when we started this project provided a simple, robust solution for the quantitation of our fluorescent images."

I would rather not call out in print the specific packages that we tried, because I cannot be confident that their failures in our hands were not our fault. However, we are now more specific with the following language, which describes our general problems and makes it clear that we are not referring to any of the packages published after 2019.

"However, we found that none of the packages available in 2019, when we started this project, provided a simple, robust solution for the quantitation of our fluorescent images. In particular, we found that packages designed for mammalian-cell images performed poorly on fission yeast."

For the reviewers information: We could not get PombeX to run, despite trying multiple hardware configurations and contacting the authors. We found ilastic far too complicated to use, particularly in terms of training. And we found that CellProfiler did a poor job of segmenting our fission yeast images.

Reviewer #6:

This manuscript provides a versatile, efficient and accurate program to automate measurements of micrograph images of fission yeast. I understand some reviewers feel the output should be compared to other software, but I do not agree. The authors have demonstrated in Fig 2 that the output is highly accurate regarding cell length and nuclear intensity. The community of biologists that study fission yeast will find this program to be extremely useful.

Specific comments

1. No page number or line numbers are provided. This makes it hard to describe comments. I will consider the abstract and Intro to be page 1.

Page numbers have been added.

2. Page 1, 9 lines from bottom. Add space to correct Tables 1and S1.

Done.

3. Page 3, text- pomBseen should be compatible with a wide range of open access…” I would like to see some more specificity here to help the reader understand which microscope systems are compatible. Can the authors provide more details.

Bio-Formats opens over 100 proprietary and open-source image formats, a list of which is available at their website <https://docs.openmicroscopy.org/bio-formats/5.3.4/supported-formats.html>. This information is now explicitly included in the manuscript. I know of no microscope system not included, but we have no objective way of stating that, beyond providing their list of supported formats.

Reviewer #7:

Briefly, the manuscript described development of an automated analysis pipeline, to measure the length and fluorescence intensity of yeast cells and their nuclei. Using programming language MATLAB for image-analysis which is widely spread and is interesting approach, however, there are some issues to be addressed in this current manuscript. My comments are below.

1. The authors should edit figures and results more comprehensive and precise. Figures seems to be insufficient of explanation of the abbreviation without consideration for readers.

We would be happy to address specific concerns. However, the request to "edit figures and results more comprehensive and precise" is too vague to be useful direction. Moreover, none of the other six reviewers have raised any concerns regarding clarity.

2. The authors should offer a more comprehensive explanation of the improvement points compared to the references presented in the Introduction section. For instance, Baybay EK et al. described the 3D segmentation of radially symmetric cells, which involved the combination of 2D brightfield segmentation with 3D morphological extrapolation using fission yeast (Reference #4). However, the authors did not provide the limitations of this research and a detailed explanation of how their pipeline improves upon the existing approach.

As we have tried hard to make clear in previous submissions, the purpose of this manuscript, and the work it describes, is not to improve on other automated analysis approaches, but rather to provide a automated approach that is as good as, but much more efficient than, manual measurement. Therefore, we have nothing to say regarding the utility of Pomegranate (Baybay EK et al.) or any other automated analysis pipeline.

3. In the Method section, the authors descripted about the images analyzed for study. A couple of strains of cells from reference (#17), however, the strains of the reference were for functional examination of the Cdc 13 and Cdc 25 expression. The authors should explain about the reason for choosing the specific strains for their study.

These strains are frequently used in the lab and serve here as simply convenient examples of the sort of fluorescent images that pomBseen can analyze. There is no specific reason for choosing them.

4. Furthermore, the authors should provide the number of image data used for validating PomBseen and more detailed calculating methods, catering to future users rather than programmers.

The four images used for validation are shown in Figure 2. It is unclear from the reviewers request what "more detailed calculating methods" they would find useful.

5. The authors should thoroughly describe the significance of employing the MATLAB program for image analysis in comparison to other methods. This aspect constitutes a key strength of the manuscript.

As explained in the Introduction, MATLAB was chosen because it is a convenient, well-supported programing environment. We did not carefully compare it to other options, so we cannot be more thorough in our comparison.

---

## [Editor Report · Decision Letter 5]

29 Aug 2023

pomBseen: An automated pipeline for analysis of Fission Yeast images

PONE-D-23-00477R5

Dear Dr.Nicholas Rhind,

We’re pleased to inform you that your manuscript has been judged scientifically suitable for publication and will be formally accepted for publication once it meets all outstanding technical requirements.

Kind regards,

Sudhir Kumar Rai, Ph.D

Academic Editor

PLOS ONE

---

## [Editor Report · Acceptance letter]

1 Sep 2023

PONE-D-23-00477R5 

pomBseen: An automated pipeline for analysis of Fission Yeast images 

Dear Dr. Rhind:

I'm pleased to inform you that your manuscript has been deemed suitable for publication in PLOS ONE. Congratulations! Your manuscript is now with our production department. 

Kind regards, 

on behalf of

Dr. Sudhir Kumar Rai 

Academic Editor

PLOS ONE